# Progress in the molecular phylogeny of *Cotesia acuminata* and *C. melitaearum* cryptic species complexes

Julian Schach[1]*, Federica Valerio[1], Anne Duplouy[1,2]*

1 Department of Organismal and Evolutionary Biology, Faculty of Biological and Environmental Sciences, The University of Helsinki, Helsinki, Finland, 2 Institute of Life Sciences (HiLIFE), The University of Helsinki, Helsinki, Finland

* julian.schach@helsinki.fi (JS); anne.duplouy@helsinki.fi (AD)

## Abstract

Cryptic species present major challenges for biodiversity and evolutionary research due to the morphological similarity and frequent lack of available genetic and ecological data. In parasitoid wasps, this is especially true for lineages with limited taxonomic and genetic sampling and unclear geographic ranges. Here we reconstruct the phylogeny of *Cotesia* wasps parasitizing Melitaeini butterflies, including two cryptic species complexes (*Cotesia acuminata* agg. and *C. melitaearum* agg.). Using a ten-gene dataset from samples collected in Europe, Asia, and North America, we inferred relationships among 22 *Cotesia* species using maximum likelihood. We also included non-Melitaeini associated *Cotesia* to assess whether the parasitoids of Melitaeini form a monophyletic group. Our analyses yielded a highly supported phylogeny, revealing four major clades, three of which included the Melitaeini associated species. This confirms that the *Cotesia* parasitizing Melitaeini butterflies are polyphyletic, likely resulting from independent host shifts across the genus. Each clade is further subdivided into subclades corresponding to the different cryptic species complexes, clarifying previously unresolved relationships. These results provide a robust framework for future studies on the evolution, ecology, and host use dynamics of *Cotesia* wasps and highlight the utility of multi-locus data for resolving phylogenies in morphologically cryptic taxa.

## Introduction

Species have historically been described using morphological data, and species relationships based on degrees of similarities of morphological characteristics. More recently, the advent of molecular techniques has refined many taxonomic/phylogenetic classifications, by allowing the study of species relationships and biodiversity with much higher resolution [1]. Cryptic species are two or more molecularly distinct species, occasionally accompanied by rapid speciation, which are or have been

**Data availability statement:** All sequence data is available from NCBI. Accession numbers: PX523132 - PX523219, PX559195 - PX559573, PX642706 - PX642796, and PX648620 - PX649011.

**Funding:** The work was funded by the Research Council of Finland to Dr. Anne Duplouy grant number 328944. The funders had no role in study design, data collection and analysis, decision to publish, or preparation of the manuscript.

**Competing interests:** The authors have declared that no competing interests exist.

erroneously classified as a single species because they are morphologically indistinguishable [2]. Such cryptic species are known to span the tree of life [3–6]. Their hidden genetic diversity has however challenged our ability to estimate the true biodiversity richness on Earth, as many remain to be discovered and characterized [7]. Furthermore, with the current biodiversity crisis, we are likely losing many such species at alarming rates even before they have been taxonomically characterized, or before their ecosystem roles were understood [8].

Parasitoid wasps make up one of the most diverse animal groups in the world [9]. They lay their eggs within or on their hosts' body, and the parasitoid larvae develop feeding on the host, until emergence of the adult wasp [10]. These parasitoids can be very efficient at killing their hosts and can act as key evolutionary pressures on their host species population growth [11] and host population dynamics. For example, it has been shown that the specialist species *Cotesia melitaearum* (Wilkinson), parasitizes only 10% of *Melitaea cinxia* (Linnaeus) caterpillars but can still cause localized extinctions within the butterfly host metapopulation [12]. Because of their narrow host ranges and highly efficient killing of their hosts, several species of parasitoids have become popular biocontrol agents in sustainable agriculture, contributing significantly to the $417 billion USD global value of insect biocontrol [13]. For example, *Cotesia ruficrus* (Haliday), parasitizes the fall armyworm (*Spodoptera frugiperda,* Lepidoptera: Noctuidae, Smith), which is contributing to food insecurity in Africa, Asia, and the Pacific [14].

Most of the world's parasitoid diversity occurs within the superfamily Ichneumonoidea, which includes the two families Ichneumonidae (Latreille) and Braconidae (Nees von Esenbeck), with approximately 60,000 described species and many more estimated [15]. Within the Braconidae, the subfamily Microgastrinae is the most diverse. For example, the genus *Cotesia* (Cameron) contains an estimated 1500–2000 species worldwide [16] of which over 300 have been named as species [17]. *Cotesia* wasps usually exhibit narrow host ranges, and closely related species often differ in their host species preferences, attacking one or two closely related lepidopteran host species [18]. For example, although previous studies suggested that seven species of *Cotesia* were generalist parasites of the Melitaeini (Newman) butterfly species found across Europe and Asia [19,20], more recent molecular analyses accompanied by rearing experiments provided evidence that the originally considered generalist parasitoid species, *Cotesia acuminata* (Reinhard) and *C. melitaearum*, were in fact comprised of numerous cryptic species, each with narrow Melitaeini butterfly host specificity [18,21].

In the complex species community of Melitaeini butterflies occurring in Catalonia in northeastern Spain (Catalan Butterfly Monitoring Scheme, CMBS), 10 local Melitaeini butterfly species were identified as hosts to at least 10 *Cotesia* (cryptic) species [18]. Unfortunately, the original barriers to geneflow between the current cryptic lineages of the wasps remain unclear, as studies exploring isolation by distance or endosymbiotic bacteria failed to explain the sympatric speciation of the parasitoid species [18,22,23]. Currently, the phylogeny of the Catalonian *C. acuminata* complex and *C. melitaearum* complex are based on 12 microsatellite loci [18], while a wider Eurasian

phylogeny of these groups combined two mitochondrial (*COI* and *ND1*) and one nuclear (*ITS2*) gene sequences [19]. As a consequence of the limited available genetic data included in these phylogenies, the relationships between *Cotesia* species still contain several unresolved nodes (i.e., nodes with low bootstrap support), and we still lack a clear description of the diversity and the speciation scenarios between *Cotesia* wasps attacking Melitaeini butterflies [18,19,21].

Here, we aimed to refine the phylogeny of *Cotesia* species associated with 16 species of Melitaeini butterflies by using ten markers and a wider sample diversity than previous studies. We analysed sequences of two mtDNA and eight nDNA markers from both freshly collected specimens from Finland, Italy, Spain, and Switzerland, as well as DNA extracts from samples collected between 1979–2002 from China, Finland, France, Hungary, Russia, Spain, Sweden, and USA [19]. With this, we expected to resolve the previously unclear nodes in the *Cotesia* phylogeny.

## Materials and methods

### Samples

This study includes *Cotesia* parasitoid wasp specimens that had emerged from the caterpillars of 16 species of Melitaeini butterflies. These include *Euphydryas aurinia* (Rottemburg), *E. aurinia davidi* (Wahlburg), *E. cynthia* (Dennis and Schiffermueller), *E. desfontainii* (Godart), *E. editha* (Boisduval), *E. maturna* (Linnaeus), *E. phaeton* (Drury), *Melitaea athalia* (Rottemburg), *M. cinxia, M. deione* (Geyer), *M. didyma* (Esper), *M.latonigena* (Eversmann), *M. parthenoides* (Keferstein), *M. phoebe* (Dennis and Schiffermueller), *M. scotosia* (Butler), and *M. trivia* (Dennis and Schiffermueller) (Table 1).

Among the *Cotesia* wasps, two cryptic species complexes are known to consist of multiple species: *C. acuminata* and *C. melitaearum*. Following previously introduced convention by Kankare *et al.* [21], we refer to these as *C. acuminata* agg. and *C. melitaearum* agg., with individual lineages denoted by species-level designations (e.g., *Sp. A* to *Sp. H*).

The Melitaeini caterpillars were collected over two time periods: 1979−2002 and 2022−2023. The DNA from early samples collected for the study of *Cotesia* wasps' phylogeny and butterfly-host specialization in earlier studies was extracted in the early 2000's. These DNA extracts represent 10 species from eight different countries, including China (N = 3 from one species), Finland (N = 5 from one species), France (N = 7 from three species), Hungary (N = 3 from one species), Russia (N = 6 from two species), Spain (N = 2 from one species), Sweden (N = 3 from one species), and USA (N = 14 from two species) (Table 1). Additionally, caterpillars from one locality in Finland (Åland) were also collected in the fall of 2022. Finally, parasitized Melitaeini caterpillars were collected in early April 2023 from five localities in Catalonia, Spain (Coll d'Estenalles, El Brull, Fígols-El Forn, Pyrenees), two localities in Switzerland (Vaud and Valais), and one locality in Italy (Aosta) (Fig 1). After collection, the caterpillars were reared in the laboratory on excess of their respective host plant, until the adult wasps emerged. Freshly emerged adult wasps were killed by placing them in a freezer for 24 hours. All samples were stored in 90% ethanol at −20°C until DNA extraction.

To test whether the *Cotesia* species parasitizing Melitaeini butterflies form a monophyletic group, we also included gene sequences retrieved from *Cotesia chilonis* (Munakata), *C. congregata* (Say), *C. glomerata* (Linnaeus), *C. ruficrus, C. typhae* (Fernández-Triana), and *C. vestalis* (Haliday), for which full genomes are publicly available from the NCBI database. These *Cotesia* species naturally parasitize species of moths (e.g., *Spodoptera* (Guenée), or *Manduca* (Hübner)) or butterflies (e.g., *Pieris* (Schrank)) [24,25]. We also included respective gene sequences from two Microgastrine species (*Microplitis demolitor* (Wilkinson), and *Microplitis mediator* (Haliday)), for which full genomes are publicly available from the NCBI database (Table 2), to use as outgroup in the phylogenetic analyses described below. We used BLAST search against the respective genomes to extract the sequences of the genes used in the phylogeny [26]. All NCBI references can be found in Table 2.

Depending on availability, we selected one to ten (on average five) parasitoid specimens per caterpillar species, and per locality sampled. As expected, the cryptic nature of the parasitoids, and the fact that some caterpillar species can host multiple species of parasitoids makes their identification difficult using only morphology. The species identity of the parasitoids collected in the early 2000's was previously confirmed through morphology, *COI,* and host associations

**Table 1. Sample summary.**

| *Cotesia* species | Butterfly host species | # of Ind. | Identified host plant | Country | Locality | Collector | Collection year |
|---|---|---|---|---|---|---|---|
| *C. acuminata Sp. A* | *Melitaea latonigena* | 3 | – | Russia | Siberia | N.W. I.H. | 1998 |
| *C. acuminata Sp. A* | *Melitaea didyma* | 5 | *Plantago lanceolata* | Spain | Coll d'Estenalles | F.V. | 2023 |
| *C. acuminata Sp. A* | *Melitaea didyma* | 5 | – | Switzerland | Conthey | Y.C. | 2023 |
| *C. acuminata Sp. B* | *Melitaea scotosia* | 3 | – | China | Peking | I.H., L.G. | 1999 |
| *C. acuminata Sp. B* | *Melitaea phoebe* | 5 | – | Italy | Aosta | Y.C. | 2023 |
| *C. acuminata Sp. K* | *Euphydryas maturna* | 1 | – | France, | Cote d'Or, Malloy, | P.J.C.R. | 1999 |
| *C. acuminata Sp. K* | *Euphydryas maturna* | 3 | – | Sweden | Lindesberg | C.U.E. | 1999 |
| *C. acuminata Sp. L* | *Melitaea athalia* | 1 | – | France | Var, St. Paul-en-Foret | P.W.C. | 1982 |
| *C. bignellii* | *Euphydryas aurinia* | 5 | – | Switzerland | La Vraconnaz | Y.C. | 2023 |
| *C. cynthiae* | *Euphydryas cynthia* | 4 | – | France | Alps, Laus de Cervieres | M.S. | 2001 |
| *C. euphydryidis* | *Euphydryas phaeton* | 7 | – | USA | Warren Co., Front Royal | N.S. | 1979 |
| *C. koebelei* | *Euphydryas editha* | 7 | – | USA | Yosemite National Park | M.S., B.W. | 2001 |
| *C. melitaearum Sp. D* | *Euphydryas desfontainii* | 5 | *Cephalaria leucantha* | Spain | Fígols-El Forn | CS | 2023 |
| *C. melitaearum Sp. D* | *Euphydryas aurinia* | 5 | *Lonicera implexa* | Spain | Cornudella de Montsant | C.S. | 2023 |
| *C. melitaearum Sp. D* | *Euphydryas aurinia* | 5 | *Lonicera implexa* | Spain | La Mussara | C.S. | 2023 |
| *C. melitaearum Sp. E* | *Melitaea deione* | 5 | *Antirhinum majus* | Spain | Les Illes | CS | 2023 |
| *C. melitaearum Sp. E* | *Melitaea deione* | 5 | *Antirhinum majus* | Spain | Poboleda | CS | 2023 |
| *C. melitaearum Sp. F* | *Melitaea didyma* | 3 | – | Hungary | Örseg | M.R.S. | 2001 |
| *C. melitaearum Sp. F* | *Melitaea didyma* | 2 | – | Spain | Cantallops | C.S. | 2002 |
| *C. melitaearum Sp. G* | *Melitaea trivia* | 10 | *Verbascum pulverulentum* | Spain | Pyrenees | C.S. | 2023 |
| *C. melitaearum Sp. H* | *Melitaea cinxia* | 8 | *Plantago lanceolata* | Finland | Åland | S.I. | 2022 |
| *C. melitaearum Sp. I* | *Melitaea athalia* | 5 | – | Finland | Åland | L.G., S.vN. | 1997 |
| *C. melitaearum Sp. M* | *Euphydryas aurinia davidi* | 3 | – | Russia | Siberia | N.W., I.H. | 1998 |
| *C. melitaearum Sp. N* | *Melitaea parthenoides* | 5 | *Plantago media* | Spain | El Brull | C.S. | 2023 |

*Cotesia* identity, host species, number of individuals, caterpillar host plant, country, locality, collector ID, and finally collection year. "-" indicates the information was unavailable.

*Collectors.* Y.C., Yannick Chitarro; P.W.C, Peter W. Cribb; C.U.E., Clae U. Eliasson; L.G., Lei Guanchung; I.H., Ilkka Hanski; S.I., Suvi Ikonen: P.J.C.R., Peter J. C. Russel: M.R.S., Mark R. Shaw; M.S., Michael Singer; N.S., Nancy Stamp; C.S., Constanti Stefansescu; S. vN., Saskya van Nouhys; N.W. Niklas Wahlberg; B.W., Brian Wee; F.V. Federica Valerio.

[19]. For the samples collected in 2022 and 2023, species identities were assigned after blasting their respective *COI* sequences against the NCBI database, and by considering their host associations, following previously established protocol [19].

## DNA extraction

For the samples collected in the early 2000's the DNA was extracted from whole insects using the NucleoSpin Tissue Kit (Macherey-Nagel) following the manufacturers protocol and eluted in 50 µL of MilliQ water (18). The samples have since been stored in the freezer (−20°C) at the University of Helsinki, Finland. For the samples collected in 2022 and 2023, the DNA was extracted destructively from each whole insects, except in two cases where we pooled damaged adult wasps that emerged from the same host caterpillar (*M. trivia*). The Macherey-Nagel NucleoSpin Tissue Kit (Düren, Germany) was used following the manufacturer's protocol. The DNA was eluted twice in 40 µL of elution buffer to increase final yield and concentration.

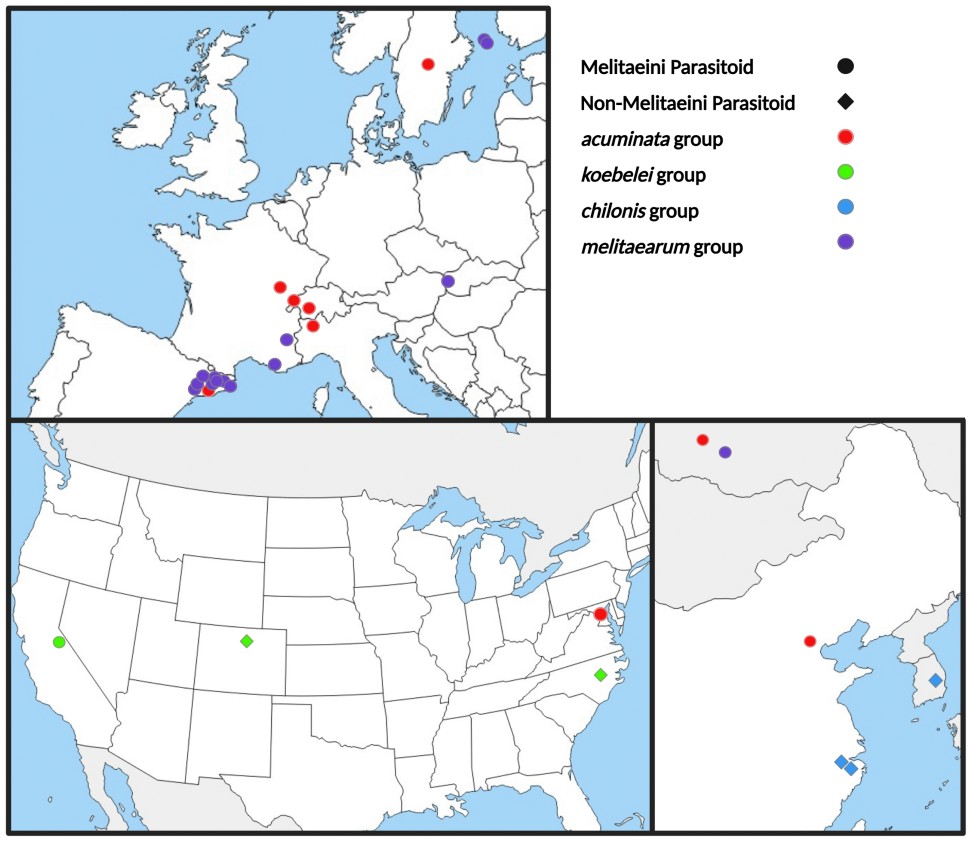

**Fig 1. Sample locations.** Sampling locations of *Cotesia* specimens used in this study. Diamonds represent species that do not parasitize Melitaeini, and circles represent species that do. Colors correspond to the different clades. Each point represents a sampling location. The African *C. typhae* sample is not included. Modified from maps freely available here: Europe (https://www.d-maps.com/carte.php?num_car=2232&lang=en), China (https://www.d-maps.com/carte.php?num_car=168&lang=en), USA (https://www.d-maps.com/carte.php?num_car=1682&lang=en). Created in BioRender. Schach, J. (2026) https://BioRender.com/futh3ay.

**Table 2. Genbank reference genomes of additional species.**

| Cotesia Species | Host | GenBank accession # | Country | Reference |
|---|---|---|---|---|
| *Cotesia chilonis* | *Chilo suppresalis* | GCA_018835575.1 | China | [27] |
| *Cotesia congregata* | *Manduca sexta* | GCA_905319865.3 | USA | [28] |
| *Cotesia glomerata* | – | GCA_020080835.1 | USA | [29] |
| *Cotesia ruficrus* | *Mythimna separata* | GCA_048537615.1 | China | – |
| *Cotesia typhae* | *Sesamia nonagrioides* | GCA_013202065.2 | Kenya | [30] |
| *Cotesia vestalis* | *Plutella xylostella* | GCA_000956155.1 | South Korea | – |
| *Microplitis demolitor* | *Pseudoplusia includens* | GCA_026212275.2 | USA | [31] |
| *Microplitis mediator* | – | GCA_029852145.1 | USA | – |

Species, host species, GenBank reference genome accession number, sampling location and reference for six species added to the *Cotesia* ingroup and *Microplitis* outgroup of the phylogenetic tree. "-" indicates the information was unavailable.

## Gene selection

Genetic markers and primer pairs designed and tested by previous studies, were selected by reviewing the available literature on *Cotesia* and Braconidae phylogenies [32–38] (Table 3). All primer pairs were tested for specificity to *Cotesia* using Primer-Blast [39]. Additionally, 16 BUSCO genes were randomly selected from the *C. congregata* genome (GCA_905319865.3) using BUSCO v 5.7.1 [40]. Using Primer-Blast, we designed an additional set of primer pairs to amplify these BUSCO genes. The best pairs were selected based on the lowest self-complimentary score given by Primer-Blast. In total, two mtDNA markers and 23 nDNA markers were tested, but only two and eight, respectively, were successfully used in this study, as 13 other nDNA primer pairs designed failed to amplify through gradient PCR (Table E in S1 Data). All primer pairs, from the literature and designed for this study, targeted genetic regions with a length ranging from 150 to 850 bp (Table 3).

## PCR

Each 10 µL PCR reaction included 1 µL of Buffer solution, 0.8 µL of DNTPs (25nM) mix, 0.5 µL of forward primers (10nM), 0.5 µL of reverse primers (10nM), 0.5 µL Mg2 reagent, 0.05 µL KAPA Taq polymerase, 6.7 µL of H$_2$0 (sterile), and 1 µL of DNA extract. The DNA extracts ranged from 90–470 ng/µL and were not normalized. The PCR reactions were initiated by incubation at 95°C for 3 min, followed by 30 cycles of 95°C for 40 s, the optimal annealing temperature of the respective primer pair (Table 2) for 40 s, and 72°C for 40 s. The PCR reaction ended with a final elongation step at 72°C for 5 min, and a final hold at 18°C.

**Table 3. Primer pairs.**

| Gene | Primer name | Sequence (5'-3') | Amplicon size | Ta (°C) | Primer source |
|------|-------------|------------------|---------------|---------|---------------|
| *COI* | LCO2198 | 5`- GGTCAACAAATCATAAAGATATTGG | 658 | 49 | [32] |
| | HCO1490 | 5`- TAAACTTCAGGGTGACCAAAAAATCA | | | |
| *16s* | 16SWb | 5`- CACCTGTTTATCAAAAACAT | 375 | 58 | [33,34] |
| | 16S outer | 5`- CTTATTCAACATCGAGGTC | | | |
| *18s* | 18s - F | 5`- CATTGGAGGGCAAGTCTGGTGCCA | 240 | 58 | [35] |
| | 18s - R | 5`- AGTAAACGTACCGGCCCTCCTCG | | | |
| *n28s rDNA* | 28s | 5`- AAGAGAGAGTTCAAGAGTACGTG | 618 | 56 | [36] |
| | 28s-PM | 5`- TAGTTCACCATCTTTCGGGTCCC | | | |
| *EF1a* | EF1A1 - F | 5`- AGATGGGYAARGGTTCCTTCAA | 418 | 56 | [37] |
| | EF1A1 - R | 5`- AACATGTTGTCDCCGTGCCATCC | | | |
| *ITS2* | NG02955 | 5`- ATGAACATCGACATTTCGAACGCACA | 153-341 | 60 | [38] |
| | AB052895 | 5`-TTCTTTTCCTCCGCTTAGTAATATGCTTAA | | | |
| *LW-Rh* | OpsFor2 | 5`-GGATGTASCTCCATTTGGTC | 432-712 | 58 | [41] |
| | Ops3'2 | 5`-AVHGATGCRACRTTCATTTTCT | | | |
| *Elob* | TCEB – F | 5`-TCTTAAAAATTCCTCCAACGCATC | 250 | 57 | This study |
| | TCEB – R | 5`-GATGCCTGCTCTTGTCCATT | | | |
| *SLD5* | SLD5 – F | 5`-CCTAAGAACGCGGTTGGAGA | 327 | 57 | This study |
| | SLD5 – R | 5`-TGAACAATATGCTGCGAGCC | | | |
| *Innexin* | INX-F | 5`-CGCTCTTTGTGTGCTTGCTT | 342 | 54 | This study |
| | INX-R | 5`-GGTACAGAAGTCGGTGAGGC | | | |

Characteristics of the primer pairs used in this study. Including the name of the marker targeted (gene), primer name, primer sequence, amplicon size, annealing temperature, and source.

The PCR products were run through a 1% agarose gel electrophoresis for 30 minutes at 100 volts, with SYBR-safe stain (Thermo Scientific, USA), and visualized under UV light. For each gel, 5 μL of 100 bp DNA ladder (#SM0321, Thermo Scientific, USA) was loaded into the first well, and each subsequent well was loaded with a mixture of 4 μL of both loading dye (bromophenol blue) and PCR product. The PCR amplification was repeated once for any failed sample, to identify potential false negative results.

### Preparation for sanger sequencing

The successfully amplified PCR products were individually cleaned using Exonuclease I (ExoI) to remove any single-stranded DNA (Thermo Scientific, USA). For each reaction 0.5 μL of ExoI was added to 5 μL of PCR product, then the samples were incubated at 37°C for 15 minutes, followed by 15 minutes at 85°C to terminate the reaction. Cleaned PCR products were sent to Macrogen Europe for Sanger Sequencing using the company Multiple Primer/Plate option.

### Sequence cleaning and alignments

The chromatograms of each sequence were manually quality-checked by using Geneious Prime 2025.01 (https://www.geneious.com). Low-quality sequences (error probability limit 0.05) and highly incomplete sequences (less than half of target length) were removed from further analysis.

We included the respective gene sequences from two Microgastrine species (*Microplitis demolitor* and *M. mediator*), as outgroup in the phylogenies described below. Additionally, six species of *Cotesia* that do not parasitize Melitaeini were added to the ingroup (Table 2). For each maker, all sequences were aligned with MUSCLE using default settings [42], and all alignments were checked by eye with AliView V1.28 [43]. The accession numbers for the sequences submitted to NCBI are PX523132-PX523219, PX559195-PX559573, PX642706-PX642796, and PX648620-PX649011 (S1 File).

### *Cotesia* phylogeny

For the phylogenetic analyses we used maximum likelihood (ML) with IQ-TREE V2.4.0 [44]. Phylogenetic analyses were conducted for: (I) all ten genes concatenated (mtDNA and nDNA), (II) eight concatenated nuclear genes, (III) two concatenated mitochondrial genes, and (IV) all genes independently. Specimens that amplified for less than half of the markers were excluded from concatenated analysis but kept for the individual gene tree analyses. Phylogenetic trees were visualized using ITOLv7.5.1 [45].

For ML analysis, we used ModelFinder [46], and the Bayesian Information Criterion (BIC) to select for the best models. The partition finding algorithm [47] was run for the concatenated data (ten-genes, nDNA, mtDNA) partitioned by genes using the MFP+MERGE option. Support for nodes was evaluated with 1000 nonparametric bootstrap replicates, where we considered values under 60 to represent unsupported nodes. Additionally, gene concordance factors (gCF) were calculated in IQ-TREE as the proportion of gene trees supporting each bipartition in the concatenated tree [48–50].

### Coalescent-based inference

To account for potential incomplete lineage sorting (ILS) we performed a quartet-based analysis in ASTRAL-III (version 5.7.8) [51], which estimates a species tree from unrooted gene trees. The input consisted of the gene trees produced by IQtree2 (see above) and a species map to account for the multiple individuals per species. We ran the ASTRAL analysis with default settings. Branch lengths were estimated in coalescent units, where shorter internal branch lengths indicated higher levels of ILS. Node support was assessed using local posterior probabilities based on quartet frequencies.

### Species delimitation

To test presence of cryptic species, we tested two methods of species delimitation. First, we used a coalescent based species delimitation with our sequence data using Bayesian Phylogenetics and Phylogeography model (BPP version

4.8.7) [52]. Then, we used a Bayesian implementation of a Poisson Tree Processes model (bPTP), which performs species delimitation on a rooted phylogenetic tree [53].

For the BPP analysis, we performed species delimitation on a guide tree (analysis A10) which uses a reversible-jump MCMC (rjMCMC) algorithm to move between species delimitation models that are compatible with a fixed tree. BPP requires prior distributions for ancestral population ($\theta$) and the age of the root ($\tau$). Because these priors have not yet been estimated for our system, we used four combinations of $\theta$ and $\tau$ to encompass a variety of demographic scenarios ranging from small populations with old divergence to large populations with recent divergence [54,55]. Specifically, we used the following combinations: $\theta = G(2, 2000)$ and $\tau = G(2,200)$, $\theta = G(2, 2000)$ and $\tau = G(2,1000)$, $\theta = G(2, 200)$ and $\tau = G(2,200)$, and $\theta = G(2, 200)$ and $\tau = G(2,2000)$. We used both the ASTRAL species tree and the ten-gene concatenated phylogeny as the guide tree to test different evolutionary hypotheses. We ran the MCMC chain for 200k generations with sampling every five generations and a burn-in of 50k generations. For the bPTP analysis, we used the bPTP web server to model species delimitation for each of our concatenated ML trees, conserving the default parameters, with outgroups removed and 500k generations.

## Results

### Parasitoid identification

The N = 42 specimens collected from 1979 to 2002 include 10 species of *Cotesia*: four cryptic species of *C. acuminata* agg., *C. cynthiae* (Nixon), *C. euphydryidis, C. koebelei* (Riley), and three cryptic species of *C. melitaearum* agg., either as previously identified by earlier studies [18,19] or by comparing their COI sequences to NCBI. The sequencing results of the 2022 and 2023 collections confirmed that these N = 68 specimens include eight species of *Cotesia*: *C. bignellii* (Marshall), two cryptic species of *C. acuminata* agg., and five cryptic species of *C. melitaearum* agg. (Table 1).

### Sequencing results

Of the 110 specimens used in this study, 41% were successfully sequenced for all 10 genes, 66% for at least 9 genes, 75% for 8 genes, 85% for 7 genes, 88% for 6 genes, and 90% for 5 genes.

### *Cotesia* phylogenies

(I) **Ten-gene dataset.** The ten-gene dataset included 110 wasp specimens (102 Melitaeini associated, and eight from NCBI). The ten-gene alignment length was 3486 bp long, of which 1498 sites (43%) represented unique site-patterns. The model partitioning algorithm revealed five partitions in the ten-gene dataset (Table A in S1 Data).

The ML analysis revealed four major clades with moderate to high bootstrap support (97, 100, 70, 97), but low to moderate gCF (50, 55, 0, 30, respectively), indicating high conflict between gene trees and the concatenated tree. The phylogeny also showed that the species attacking Melitaeini butterflies are polyphyletic (Fig 2). The node containing the *acuminata*, *vestalis*, and *melitaearum* groups was poorly supported (bootstrap < 60, gCF = 0).

- **The *koebelei* group** contained *C. koebelei, C. congregata* and *C. glomerata* parasitoid wasps. *Cotesia congregata* and *C. glomerata* do not use Melitaeini butterflies as hosts, but *C. koebelei* wasps parasitize the Melitaeini species *E. editha*. The nodes in this clade were highly supported (bootstrap = 91 and above)

- **The *acuminata* group** was formed by the *C. acuminata* complex, *C. bignellii,* and *C. euphydryidis* parasitoids from Melitaeini butterflies. *Cotesia euphydryidis* was placed at the base of the clade. The next division was between *C. bignellii* and *C. acuminata* agg. The clade revealed *C. acuminata* agg. as polyphyletic, where *Sp. B* and *K* cluster together and are sister taxa to *C. bignelli*. The divergence between *C. acuminata Sp. A* and *C. bignelli, C. acuminata Sp. B* and *K* was well supported (bootstrap = 91) but the relationship between *C. bignelli, C. acuminata Sp. B* and *K* was not.

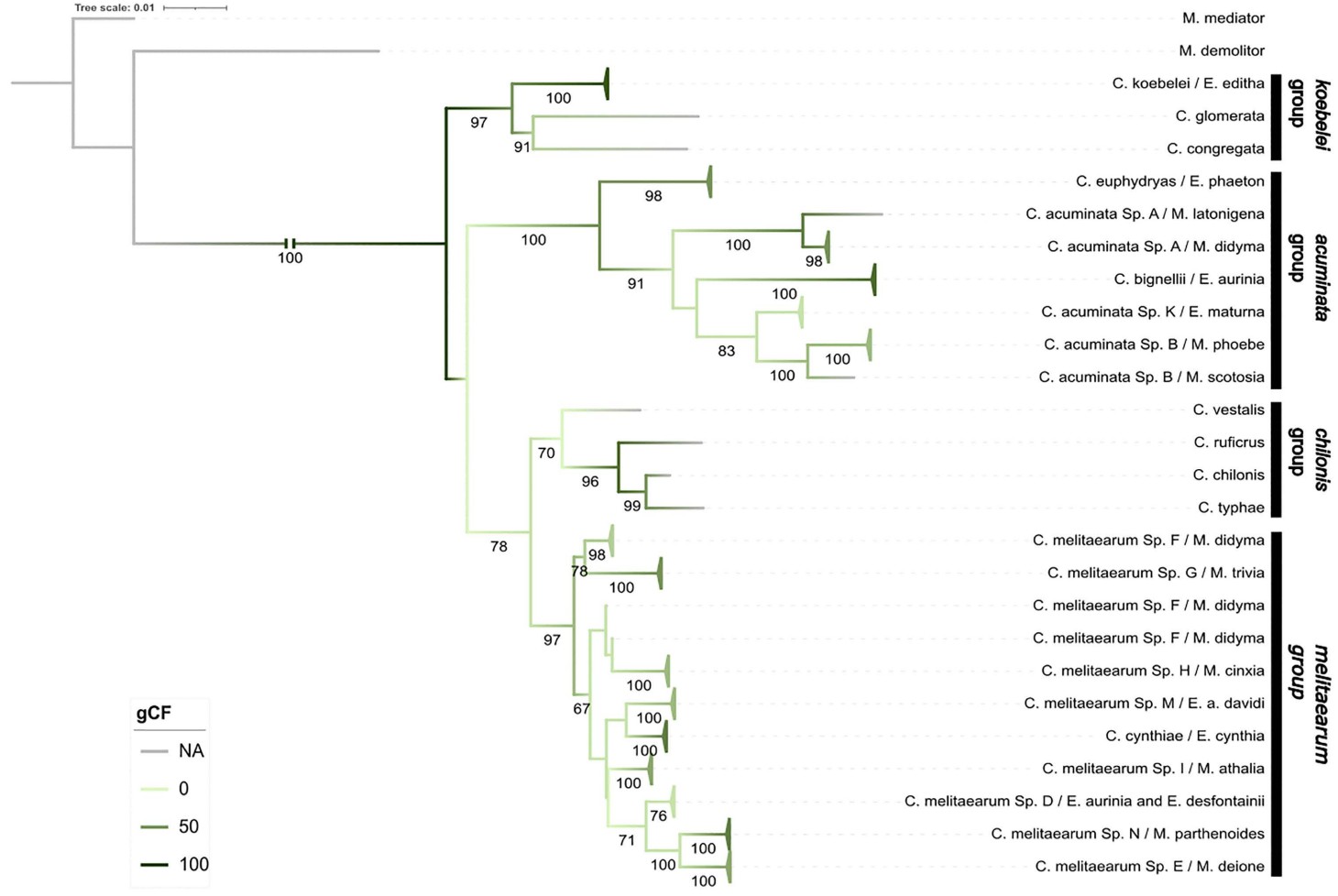

**Fig 2. Ten-gene maximum likelihood phylogeny.** Maximum likelihood phylogeny of *Cotesia* species parasitizing different Melitaeini host species and their relatives based on the ten concatenated mitochondrial and nuclear genes (16s, *18s, 28s, COI, EF1a, INX, ITS2, LW-Rh, SLD5,* and *Elob*). The specimens are labelled by the names of the *Cotesia* species and Melitaeini host caterpillar species. Bootstrap support values (1000 replicates) are indicated for supported branches (≥60). The branches are colored according to gene concordance factor (gCF). Branches corresponding to the same species were collapsed into triangles for visual clarity.

- **The *chilonis* group** contained *C. chilonis, C. ruficrus, C. typhae,* and *C. vestalis* parasitoids, which all parasitize non-Melitaeini hosts. The relationships in this clade were well supported.

- **The *melitaearum* group** contained *C. melitaearum* agg. and *C. cynthiae,* which parasitize Melitaeini butterflies. Ten subclades were revealed, represented by the individual cryptic species and *C. cynthiae* with varying support (bootstrap 55–100). Here, *C. melitaearum* agg. formed a paraphyletic group. Most of the individual species within this clade grouped together as monophyletic, except *C. melitaearum Sp. F*. For this particular species, our phylogeny placed the specimens from Spain in a sister group to *C. melitaearum Sp. G* also from Spain and the specimens from Hungary were paraphyletic and sister to the *C. melitaearum Sp. H* from Finland. Additionally, *C. melitaearum Sp. D* was grouped together and not clustered by host species.

**(II) Nuclear-gene dataset.** The nuclear-gene dataset included 112 wasp specimens (104 Melitaeini-associated, and eight from NCBI). The concatenated alignment was 2413 bp long, of which 1035 sites (43%) represented unique site-patterns. The model partitioning algorithm revealed two partitions in the nuclear-gene dataset (Table B in S1 Data).

The ML analysis produced a moderately supported phylogeny that recovered the same four clades as the ten-gene dataset with varying bootstrap support (100, 96, 51, 84) and low gCF (57, 50, 0, 12.5). The placements of the *acuminata* and *koebelei* groups were switched when compared to the ten-gene concatenated tree, however the node containing the *koebelei, chilonis,* and *melitaearum* groups was poorly supported (bootstrap < 60, gCF = 12.5) (Fig 3). Additionally, there were some differences within the *acuminata* and *melitaearum* groups between the two trees.

- **The *acuminata* group** recovered *C. acuminata* agg. as monophyletic, although poorly supported (bootstrap = 39).

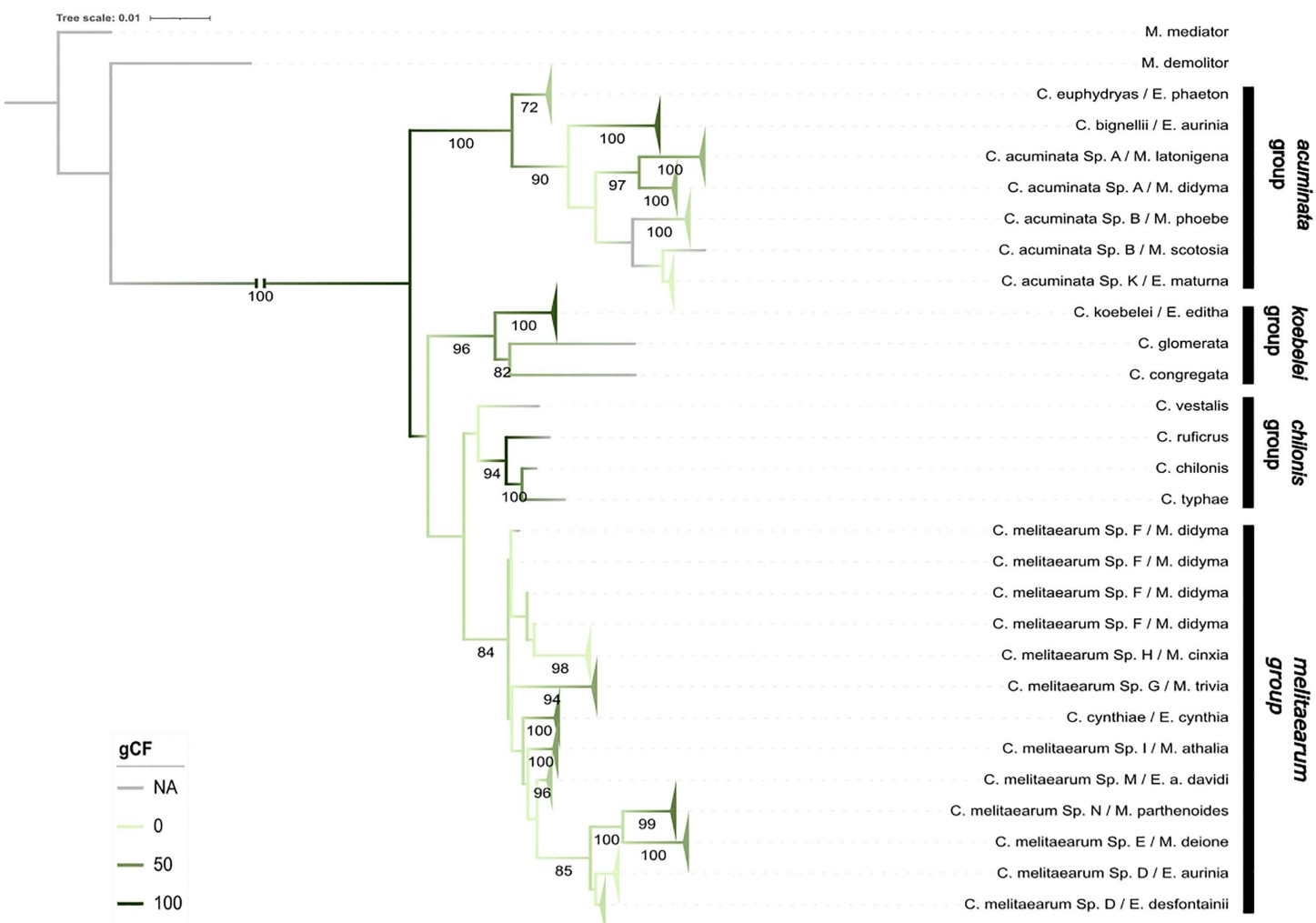

**Fig 3. Nuclear maximum likelihood phylogeny.** Maximum likelihood phylogeny of *Cotesia* species parasitizing different Melitaeini host species and their relatives based on the nuclear genes (*18s, 28s, EF1a, INX, ITS2, LW-Rh, SLD5,* and *Elob*). The specimens are labelled by the names of the *Cotesia* species and Melitaeini host caterpillar species. Branches corresponding to the same species were collapsed for visual clarity and represented by triangles. Bootstrap support values (1000 replicates) are indicated for supported branches (≥60). The branches are colored according to gene concordance factor (gCF). Branches corresponding to the same species were collapsed into triangles for visual clarity.

- The *melitaearum* group showed some discordance from the ten-gene phylogeny. For example, *C. melitaearum Sp. F* was grouped together with *C. melitaearum Sp. H* however as paraphyletic, thus resulting in nine subclades. Many of which were well supported, but the relationships between them were not and have low gCF values. Additionally, *C. cynthiae* and *C. melitaearum Sp. M* were no longer sister groups.

(III) **Mitochondrial-gene dataset.** The mitochondrial-gene dataset included the partial sequences of the *16s* and *COI* genes from 92 specimens (86 Melitaeini associated, six from NCBI). The mitochondrial alignment was 1072 bp long, of which 451 (42%) represented unique site-patterns. The model partitioning algorithm revealed two partitions in the mitochondrial gene dataset, which correspond to the two markers (Table C in S1 Data). No specimens from *C. euphydryidis* and *C. melitaearum Sp. F* from Hungary amplified for either mitochondrial marker, and no sequences were found from *C. typhae* and *C. vestalis* genomic projects, thus they were missing from the mitochondrial phylogenetic analysis. The ML analysis revealed the same four major clades as the ten-gene and nuclear gene phylogenies, however with varying bootstrap support (62, 91, 98, 58) and gCF (50, 100, 50, 100) (Fig 4). The relationships within the *koebelei* and *chilonis* groups were congruent to the ten-gene and nuclear datasets, however, two species were missing from the *chilonis* group.

- The *acuminata* group recovered the *C. acuminata* complex as monophyletic, similar to the nuclear dataset, however the support was low (bootstrap<60).

- The *melitaearum* group formed eight well supported subgroups corresponding to the different cryptic species (bootstrap 76–99) and maintained the paraphyletic nature of *C. melitaearum* agg. however, many of the relationships between clades were unsupported (bootstrap<60).

(IV) **Individual genes.** All genes were also analysed individually using ML to estimate gene concordance factors, and to build a coalescent based species tree. The substitution models for the ML analyses are listed in Table D in S1 Data. Rather expectedly, the individual gene phylogenies had much lower branch supports than either of the other phylogenies, likely because of the small nucleotide sample size included in each analysis. Alone, the individual gene trees are not phylogenetically informative. The major clades were only recovered in the *16s*, *COI*, and *SLD5* ML trees, but the relationships within the clades, particularly within the *melitaearum* group, remain incongruent (S7–S16 Figs).

## Coalescent-based inference

Individual gene trees were also analysed together under the multispecies coalescent using ASTRAL-III. While the ASTRAL analysis recovered the same major clades as the concatenated ML phylogenies, the relationships within the clades were more often weakly supported and topologically incongruent from the concatenated analyses. The coalescent analysis placed the *acuminata* group as the ancestral clade and did not propose the *chilonis* group as monophyletic, as *C. vestalis* diverges from the split between the *chilonis* and *melitaearum* groups. These differences might be caused by the lower amount of sequence data for *C. vestalis* (only five loci were available for that particular species). Overall, internal nodes within the coalescent tree exhibited low support (LPP<0.7) with short branches, indicating substantial gene tree discordance (S1 Fig).

## Species delimitation

The Bayesian species delimitation provided strong support for the *a priori* defined species, and the results were robust to the guide tree, and priors on population size (θ) and root age (τ) (S2 and S3 Fig). Each of our cryptic species had high posterior support (1.00), however we observed some over-splitting, for example, *C. acuminata Sp. B* and *C. melitaearum Sp. D* from different host species/populations were recovered as independent species. The lowest supported split was between *C. chilonis* and *C. typhae*, however they were still maintained as separate species. The bPTP analysis of the

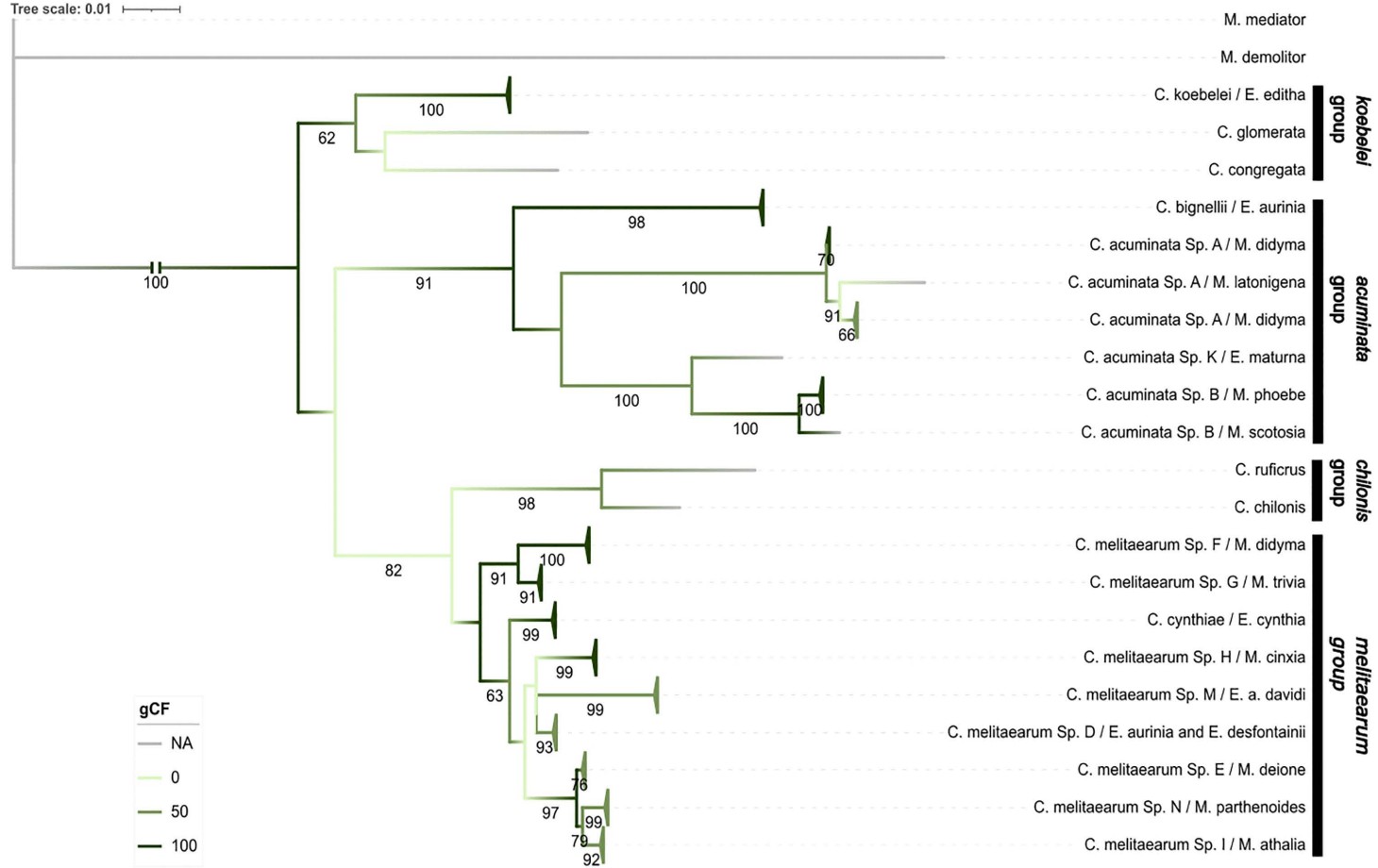

**Fig 4. Mitochondrial maximum likelihood phylogeny.** Maximum likelihood phylogeny of *Cotesia* species parasitizing different Melitaeini host species and their relatives based on the mitochondrial genes (*16s* and *COI*). The specimens are labelled by the names of the *Cotesia* species and Melitaeini host caterpillar species. Branches corresponding to the same species were collapsed for visual clarity and represented by triangles. Bootstrap support values (1000 replicates) are indicated for supported branches (≥60). The branches are colored according to gene concordance factor (gCF). Branches corresponding to the same species were collapsed and represented by triangles for visual clarity.

concatenated phylogenies supported the presence of cryptic species and distinct clades (S4–S6 Figs). Although the bPTP never lumped together cryptic species into a single species, this analysis suggested more over-splitting of cryptic species, than previous analyses. For example, the three *C. melitaearum Sp. I* individuals were categorized as separate species in the bPTP analysis.

## Discussion

Resolving species phylogenies is essential for studying trait evolution, diversification, and ecological interactions, but can remain challenging in non-model cryptic species, due to limited access to sampling and molecular resources. By designing new primers and increasing the number of available molecular markers from two to ten, we were able to produce the most robust phylogeny of the *Cotesia* species parasitizing Melitaeini butterflies to date. This improved phylogeny informs on previously hidden species diversity within this genus and is an important resource for future research on Braconidae population genetics, evolutionary dynamics and ecology.

Our analyses revealed that all specimens group in four highly supported clades (*i.e., koebelei group, acuminata group, chilonis group, and melitaearum group*), each subdivided into distinct subclades of (cryptic) species. While our results showed strong support for the species groups, the relationships between them are still unresolved, similar to other *Cotesia* multi-gene phylogenies [19,56]. By including genetic sequences from *Cotesia* species that do not attack Melitaeini butterflies, we confirm that the Melitaeini-specialized species are polyphyletic, appearing in three of the four main *Cotesia* clades. This suggests that *Cotesia* wasps may have shifted hosts multiple times throughout their evolutionary history. Such patterns of host shifts have been observed in other parasitoid lineages, including Ichneumonoidea [57], and parasitoids of leaf miners [58], indicating that parasitoid evolution often involves ecological switching rather than strict co-divergence with hosts.

The presence of cryptic species was confirmed using two methods of species delimitation (i.e., BPP and bPTP analyses). The BPP model strongly supported the predefined species taxonomy, with the lowest supported split being between *C. chilonis* and *C. typhae* that do not attack Melitaeini butterflies. However, this may be due to the presence of only one individual for these species, making it impossible to estimate within species population parameters [52]. The coalescent based delimitation tended to over-split especially when a single species was differentiated by their host species, likely capturing population structure as distinct species, and which can be exacerbated by the presence of speciation with gene flow [59]. Similarly, the bPTP over-split the species taxonomy, Which is maybe not surprising as this method relies entirely on branch length to distinguish between intra- and interspecific variation, and tends to over-split species particularly when there is geneflow between species and deep ILS [60], which is likely the case for these rapidly diverging cryptic species. Additionally, missing data in the concatenated alignments could have distorted branch lengths furthering over-splitting in bPTP [61].

Many of the Melitaeini-associated *Cotesia* species occur in sympatry, yet the drivers of radiation and barriers to gene flow remain unclear [18,22]. The butterfly host evolution could, at least partially, affect the divergence between close *Cotesia* lineages and cryptic species. Indeed, while most *C. acuminata* species parasitize *Melitaea* butterflies, the ancestral species parasitize *Euphydryas* caterpillars, which have diverged from *Melitaea* 42.68 MYA [62]. Similarly, in the *koebelei* group, the *C. koebelei* species parasitizes *E. editha* and has also been tentatively reported on *Melitaea* [63], this may suggest the parasitoid is a rare generalist able to attack both butterfly genera. Alternatively, if a preference would be detected between the *C. koebelei* lineages parasitizing *Euphydryas versus Melitaeini* butterflies, the host difference could simply indicate the cryptic diversity in this species complex. The question of which of these two hypotheses is the most plausible could potentially be resolved through future intensive sampling of North American Melitaeini butterflies and their *Cotesia* parasitoids.

In contrast, the *melitaearum* group shows an opposite pattern to the *acuminata* group: with the ancestral species parasitizing *Melitaea*, followed by a shift to *Euphydryas*. This may reflect a retained ability to use *Euphydryas, or* recolonization event of *Euphydryas*, the possible host of the ancestor shared between the *melitaearum* group and the *acuminata* group. In contrast, species from the *chilonis* group are not known to use Melitaeini butterflies but span a wider range of Lepidoptera host taxa. For example, *C. vestalis* parasitizes *Plutella xylostella,* (Linnaeus) (Yponomeutoidea), a lineage near the base of the Lepidopteran tree [64], while more recently diverged *Cotesia* in this clade parasitize more recently diverged groups such as Pyraloidea, (Latreille) and Noctuidae, (Latreille).

Although the relationships within and between cryptic species are better supported in our phylogeny compared to previous studies [19], comparisons of the tree topologies, concordance factors, and ASTRAL suggested conflicts between nuclear and mitochondrial reconstructions and gene trees. The discordance within clades may reflect biological processes such as geneflow between lineages or rapid radiation resulting in incomplete lineage sorting [65,66]. The non-random absence of mitochondrial data for several taxa in two clades can bias phylogenetic inference, especially when relatively few markers are used [61], and could partially affect the differences observed between datasets used here. Furthermore, the ten loci and missing data may not provide enough signal for a comprehensive ASTRAL analysis, resulting in the poorly supported species ASTRAL tree, which exhibit very short branches.

While we present an improved *Cotesia* phylogeny, future work may require additional genetic data, broader geographic sampling, and inclusion of other species to fully evaluate the evolutionary history of the genus *Cotesia*. Although whole genome sequencing approaches might provide more detailed information on hybridization and introgression events, or on lineage sorting in the genus *Cotesia,* the integration of new experimental and field data on host fidelity (as in Kankare et al [18]) and/or reproductive isolation would reveal elements of the ecological barriers that have facilitated the speciation of *Cotesia* wasps. Many parasitoid wasp females preferentially oviposit on the same host species from which they emerged, reinforcing such reproductive barriers and potentially promoting speciation [67–69]. Alternatively, the use of large network-based approaches could offer useful frameworks to further investigate host-parasitoid coevolution [70], testing hypotheses such as adaptive radiation [71], or host use oscillations [72]. Our improved *Cotesia* phylogeny offers a great first step towards the evaluation of *Cotesia* species diversity. Combined with a clearer understanding of other aspects of *Cotesia's* ecology, such as ecological niches and host range breadth, this data will support conservation of its cryptic diversity, as many species with narrow host ranges may face elevated extinction risk. Our study questions the need to revise the taxonomy of *Cotesia* to highlight the cryptic diversity, and the possible introduction of new sub-genera by taxonomists.

## Conclusion

Our study provides a robust phylogenetic framework for *Cotesia* parasitizing Melitaeini butterflies, clarifying uncertainties in their evolutionary relationships and cryptic diversity. The results show that the species attacking Melitaeini are paraphyletic, grouping in three of the four major clades, with indications of independent host shifts. Although our sampling does not allow for inference of which lineage first adapted to Melitaeini hosts, the confirmation of distinct cryptic species in *C. acuminata* agg. and *C. melitaearum* agg. highlights the extent of hidden diversity in this system. Together, these findings establish a foundation for future taxonomic revisions, comparative genomic studies, and exploration of the ecological barriers that initiated and maintain their reproductive isolation.

## Supporting information

**S1 Data. Tables. Table A. Ten-gene partitioning.** Ten-gene dataset partitions determined by partitioning algorithm, and best fit model as determined by ModelFinder using BIC. **Table B. Nuclear partitioning.** Nuclear gene dataset partitions determined by partitioning algorithm, and best fit model determined by ModelFinder using BIC. **Table C. Mitochondrial partitioning.** Mitochondrial-gene dataset partitions determined by partitioning algorithm, and best fit model determined by ModelFinder using BIC. **Table D. Substitution models.** Substitution models used for individual genes, determined by ModelFinder, and selected based on BIC. **Table E. Primer pairs.** Characteristics of the additional primer pairs created for this study. Includes the marker targeted (gene), primer name, primer sequence, amplicon size.
(DOCX)

**S1 File. Sequence information.** Species ID, sample ID, approximate GPS location, Host ID, and genbank accession number for each sample sequenced in this study. ¨-¨ represents the individual was not sequenced for that locus.
(XLSX)

**S1 Fig. Coalescent based phylogeny.** Species tree inferred using ASTRAL-III relatives based on the ten mitochondrial and nuclear gene trees (16s, *18s, 28s, COI, EF1a, INX, ITS2, LW-Rh, SLD5,* and *Elob*). Branch lengths are expressed in coalescent units. Node support values represent local posterior probabilities (LPP).
(TIF)

**S2 Fig. BPP ASTRAL guide tree.** Bayesian Phylogenetics and Phylogeography (BPP) results using the Coalescent based species tree estimations. Speciation probabilities from BPP for each node are shown in boxes: Top, $\theta = G(2, 2000)$

and τ = G(2,200); second, θ = G(2, 2000) and τ = G(2,1000); third θ = G(2, 200) and τ = G(2,200); bottom, θ = G(2, 200) and τ = G(2,2000).
(TIF)

**S3 Fig. BPP ML guide tree.** Bayesian Phylogenetics and Phylogeography (BPP) results using the ML tree estimations. Speciation probabilities from BPP for each node are shown in boxes: Top, θ = G(2, 2000) and τ = G(2,200); second, θ = G(2, 2000) and τ = G(2,1000); third θ = G(2, 200) and τ = G(2,200); bottom, θ = G(2, 200) and τ = G(2,2000).
(TIF)

**S4 Fig. Ten-gene dataset bPTP species delimitation tree.** Ten-gene maximum likelihood phylogeny showing bPTP results. Red clades represent putative species.
(TIF)

**S5 Fig. Nuclear dataset bPTP species delimitation tree.** Nuclear DNA maximum likelihood phylogeny showing bPTP results. Red clades represent putative species.
(TIF)

**S6 Fig. Mitochondrial dataset bPTP species delimitation tree.** Mitochondrial DNA maximum likelihood phylogeny showing bPTP results. Red clades represent putative species.
(TIF)

**S7 Fig. *16s* gene tree.** Maximum likelihood phylogeny of *Cotesia* species parasitizing different Melitaeini host species and their relatives based on the mitochondrial gene *16s*. The specimens are labelled by the names of the *Cotesia* species and Melitaeini host caterpillar species. Bootstrap support values (1000 replicates) are indicated for supported branches (≥60).
(TIF)

**S8 Fig. *18s* gene tree.** Maximum likelihood phylogeny of *Cotesia* species parasitizing different Melitaeini host species and their relatives based on the mitochondrial gene *18s*. The specimens are labelled by the names of the *Cotesia* species and Melitaeini host caterpillar species. Bootstrap support values (1000 replicates) are indicated for supported branches (≥60).
(TIF)

**S9 Fig. *28s* gene tree.** Maximum likelihood phylogeny of *Cotesia* species parasitizing different Melitaeini host species and their relatives based on the mitochondrial gene *28s*. The specimens are labelled by the names of the Cotesia species and Melitaeini host caterpillar species. Bootstrap support values (1000 replicates) are indicated for supported branches (≥60).
(TIF)

**S10 Fig. *COI* gene tree.** Maximum likelihood phylogeny of *Cotesia* species parasitizing different Melitaeini host species and their relatives based on the mitochondrial gene *COI*. The specimens are labelled by the names of the *Cotesia* species and Melitaeini host caterpillar species. Bootstrap support values (1000 replicates) are indicated for supported branches (≥60).
(TIF)

**S11 Fig. *EF1a* gene tree.** Maximum likelihood phylogeny of *Cotesia* species parasitizing different Melitaeini host species and their relatives based on the mitochondrial gene *EF1a*. The specimens are labelled by the names of the *Cotesia* species and Melitaeini host caterpillar species. Bootstrap support values (1000 replicates) are indicated for supported branches (≥60).
(TIF)

**S12 Fig. *INX* gene tree.** Maximum likelihood phylogeny of *Cotesia* species parasitizing different Melitaeini host species and their relatives based on the mitochondrial gene *INX*. The specimens are labelled by the names of the *Cotesia* species and Melitaeini host caterpillar species. Bootstrap support values (1000 replicates) are indicated for supported branches (≥60).
(TIF)

**S13 Fig. *ITS2* gene tree.** Maximum likelihood phylogeny of *Cotesia* species parasitizing different Melitaeini host species and their relatives based on the mitochondrial gene *ITS2*. The specimens are labelled by the names of the *Cotesia* species and Melitaeini host caterpillar species. Bootstrap support values (1000 replicates) are indicated for supported branches (≥60).
(TIF)

**S14 Fig. *LW-Rh* gene tree.** Maximum likelihood phylogeny of *Cotesia* species parasitizing different Melitaeini host species and their relatives based on the mitochondrial gene *LW-Rh*. The specimens are labelled by the names of the *Cotesia* species and Melitaeini host caterpillar species. Bootstrap support values (1000 replicates) are indicated for supported branches (≥60).
(TIF)

**S15 Fig. *SLD5* gene tree.** Maximum likelihood phylogeny of *Cotesia* species parasitizing different Melitaeini host species and their relatives based on the mitochondrial gene *SLD5*. The specimens are labelled by the names of the *Cotesia* species and Melitaeini host caterpillar species. Bootstrap support values (1000 replicates) are indicated for supported branches (≥60).
(TIF)

**S16 Fig. *Elob* gene tree.** Maximum likelihood phylogeny of *Cotesia* species parasitizing different Melitaeini host species and their relatives based on the mitochondrial gene *Elob*. The specimens are labelled by the names of the *Cotesia* species and Melitaeini host caterpillar species. Bootstrap support values (1000 replicates) are indicated for supported branches (≥60).
(TIF)

## Acknowledgments

Thanks to the members of the ISEE lab and life-history evolution group at the University of Helsinki for the many discussions and input. Thank you to Dr. Victoria Twort for her input on the manuscript. Thanks to everyone involved in the sample collection, including Constantí Stefanescu and his team in Spain, Yannick Chittaro in Switzerland, and in Finland Suvi Ikonen and Krista Raveala, along with those involved in the Åland butterfly survey.

## Author contributions

**Conceptualization:** Anne Duplouy.

**Data curation:** Julian Schach.

**Formal analysis:** Julian Schach.

**Funding acquisition:** Anne Duplouy.

**Investigation:** Julian Schach.

**Methodology:** Julian Schach, Federica Valerio.

**Project administration:** Anne Duplouy.

**Supervision:** Federica Valerio, Anne Duplouy.

**Writing – original draft:** Julian Schach.

**Writing – review & editing:** Julian Schach, Federica Valerio, Anne Duplouy.

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
