## [Decision Letter · Decision Letter 0]

10 Mar 2026

PONE-D-26-02522Progress in the molecular phylogeny of Cotesiaacuminata and C. melitaearum cryptic species complexesPLOS One

Dear Dr. Schach,

Thank you for submitting your manuscript to PLOS ONE. After careful consideration, we feel that it has merit but does not fully meet PLOS ONE’s publication criteria as it currently stands. Therefore, we invite you to submit a revised version of the manuscript that addresses the points raised during the review process.

We look forward to receiving your revised manuscript.

Kind regards,

Petr Heneberg

Academic Editor

PLOS One

Journal Requirements:

“The work was funded by the Research Council of Finland to Dr. Anne Duplouy grant number 328944”

6. Please amend either the title on the online submission form (via Edit Submission) or the title in the manuscript so that they are identical.

Reviewers' comments:

Reviewer's Responses to Questions

**Comments to the Author**

1. Is the manuscript technically sound, and do the data support the conclusions?

Reviewer #1: Yes

Reviewer #2: Partly

Reviewer #3: Partly

Reviewer #4: Partly

2. Has the statistical analysis been performed appropriately and rigorously? 

Reviewer #1: Yes

Reviewer #2: No

Reviewer #3: No

Reviewer #4: Yes

3. Have the authors made all data underlying the findings in their manuscript fully available?

Reviewer #1: No

Reviewer #2: Yes

Reviewer #3: Yes

Reviewer #4: No

4. Is the manuscript presented in an intelligible fashion and written in standard English?

Reviewer #1: Yes

Reviewer #2: Yes

Reviewer #3: Yes

Reviewer #4: Yes

5. Review Comments to the Author

Reviewer #1: The paper presents a multi-gene phylogeny for braconid wasps in genus Cotesia with improved resolution. It is suitable for publication in Plos One after a minor revision.

In addition to the changes made in the attached word document, I have the following five main concerns:

1) the abbreviation agg. should not be italicized, as it is not part of the scientific name of the species. The paper that the authors cite for the use of this term (Kankare et al 2005) also does not italicize the term.

2) Table 1 does not include GenBank numbers for each individual analysed, and it is unclear which gene was sequenced for which sample. From Table 1 it can be inferred that the authors had a total of about 105 individuals (plus some from NCBI). A new supplementary table is needed with a list of these 105 individuals, their collection data and GPS coordinates (even if approximate), followed by columns for each of the 10 genes and the GenBank accessions for each individual clearly indicated.

3) The authors mention sympatric occurrence of some of the Cotesia species, however without a map it is difficult to understand the extent of such sympatry. A new separate figure is needed with a map marking the geographic locations of the studied samples. Different symbols and colors should be used for the dots on the map to show: a) samples belonging to the four main clades, and b) whether or not they parasitize Melitaeini.

4) It is unclear how the authors decided to separate and highlight their four clades. For example, Clades C and D could be considered a single clade … I suggest that the authors conduct a basic Species Delimitation analysis (such as ASAP or PTP) or something similar, at least on their final phylogeny, to support their clade/species designations and conclusions.

5) The authors make no suggestions to improve the systematics of Cotesia. Could the four main clades perhaps be considered subgenera? or species-groups? I suggest the authors to include at least a paragraph about this in their discussion.

Reviewer #2: The authors undertake a study to investigate phylogenetic relationships among Cotesia species parasitizing the caterpillars of Melitaeini butterflies. They do this using 102 fresh samples of Cotesia combined with and eight species for which data was available on GenBank. Their outgroup consisted of two Microplitis species. They generated data from 10 gene fragments (8 nuclear, 2 mitochondrial) for their taxa and analyzed them in a concatenated framework and by individual gene using maximum likelihood.

The authors show the Cotesia associated with Melitaini are non-monophyletic and the existence of multiple forms within the currently recognized species C. acuminata and C. melitaerum.

My main critique of the study is the limited approach to phylogenetic analysis and the unsupported statements regarding the production of a “robust” phylogenetic framework.

The authors only use a single analytical approach to phylogenetic inference – maximum likelihood (ML) as implemented in IQ-Tree. While the authors do investigate the influence of combining mitochondrial genes with nuclear genes, and the results of individual gene analyses, these can merely identify contention within the data themselves rather than the effects of different analytical approaches. Most modern phylogenetic studies at least incorporate some form of coalescent approach in order to account for gene tree/species tree discordance and incomplete lineage sorting directly. The authors identify the presence of gene tree discordance through analysis of their individual loci. However, they do not produce a coalescent-based tree to help resolve this. A common software package for this is ASTRAL (https://github.com/smirarab/ASTRAL) which makes use of summary coalescence – analyzing a priori produced gene trees to infer a species tree. The authors have already done most the work producing individual ML trees and would simply have to provide these to ASTRAL to produce a species tree.

Additionally, given the highly tractable size of their data, they could also perform Bayesian inference (BI) using MrBayes (https://nbisweden.github.io/MrBayes/) or Beast (https://www.beast2.org/tutorials/).

A highly robust phylogenetic framework would be obtained from reciprocal illumination of results generated using different analytical approaches such as ML, BI, and ASTRAL. If these three fundamentally different approaches produce similar trees, then the data and inferred evolutionary history is robust to analytical approach.

Finally, the results the authors did obtain do not provide robust resolution across their datasets. An ultrafast bootstrap value of ≥95 is considered strong support (see reference 38), anything below this is not strongly supported. Figure 1, Clade (B(C+D)) shows low bootstrap support 77 (the nest node is illegible), Figure 2, the same Clade (B(C+D)) has similarly low bootstrap support – 82, then 87. Finally, figure 3, the analysis of mitochondrial genes alone recovers a different topology where B moves out and A moves in to clade. Bootstrap value is notably missing for this node in Figure 3. Combined these suggest that both within analysis support for much of their recovered topology is not strong, and the same topologies are not recovered across all datasets. So, while the recovery of species-group clades is consistent, relationship among said clades and thus the backbone of the phylogeny is not robust.

I recommend the authors include additional analytical approaches, at the minimum ASTRAL. Ideally, BI and ASTRAL.

The authors indicate they suspect cryptic species, they could also perform a species delimitation analysis using BPP (https://github.com/bpp/bpp) and their mitochondrial data as well as their preferred tree as a guide tree to see how many of these potential cryptic species-level units are supported in a multispecies coalescent framework. Currently they have identified consistent splits in ML trees, but the nature of which of these splits are species-level is unclear, particularly for C. melitaearum which shows less structure than C. acuminata. Species delimitation may support C. melitaerum samples as being part of a single species relative to other taxa in the tree like C. acuminata.

On lines 90 – 92 the authors state they want to resolve basal nodes of Cotesia, suggesting the broader phylogeny is of interest to them. However, based on the rest of the study it really seems the true focus is that discussed in 85 – 87. Therefore, I suggest removing this line as the broader phylogeny of Cotesia was really not a focus.

The authors fail to reference Michel-Salzat & Whitfield 2004 which appears to be an important study on the broader Cotesia phylogeny. This should be referenced.

Michel-Salzat, A. and Whitfield, J.B. (2004), Preliminary evolutionary relationships within the parasitoid wasp genus Cotesia (Hymenoptera: Braconidae: Microgastrinae): combined analysis of four genes. Systematic Entomology, 29: 371-382. https://doi.org/10.1111/j.0307-6970.2004.00246.x

Finally, a note that the discussion should probably be narrative and not include bullet points. There are also a couple issues with incorrectly formatted references (Line 56 & 83).

Reviewer #3: General

The submitted manuscript by Schach et al. uses a phylogenetics approach to investigate species complex relationships in a parasitoid wasp group. The analyses use 10 loci (eight nuclear and two mitochondrial) and a concatenated maximum likelihood approach. The results showed that the parasitoid wasp lineages were paraphyletic, resulting from numerous shifts in hosts.

Many recent/current phylogenomic studies utilize Illumina NGS data to isolate hundreds to thousands of loci. While that is common, not every study needs that level of data, especially projects that are long-standing or have a small budget. I am not going to ask for that here, since that seems unrealistic. However, I would say that many of the current analysis approaches used in those study should be at least considered here. Those include a coalescent-based approach such as Astral. Ten loci is not very many, yet there may be enough signal there. The concatenated approach is likely the better option, as shown here, but more work should be done to show concordance/discordance in the data used. Approaches such as PhyParts or gene concordance factors would show how many of the individual genes support different nodes in the concatenated tree. Bootstrap support, which tends to be inflated in phylogenomic data sets, is primarily a metric of confidence in the topology, not so much in the level of support of the data (either genes or sites) of the topology. In addition to bootstrap support, it would be useful to see how site concordance factors support the concatenated topology.

I have a lot of specific comments to follow, many of these involve the presentation of the study. There are numerous grammatical or punctuation mistakes throughout. A careful revision is needed. Towards the end of the discussion the formatting of the intext citations completely changes from the rest of the manuscript. I also have many comments involving being explicit. Instead of saying “a few”, say how many. When saying that mito and nuclear loci have been used previously, say which ones. These things will help nonexperts in parasitoid wasp evolution understand the manuscript. Overall, there is nothing wrong with the underlying data. This is a great foundation for a solid manuscript, but some reworking of analyses and revision of the main text is needed before publication. One specific example is the talk of species or cryptic species; a monophyletic group in a phylogenetic tree may not be enough evidence to assign species. There are numerous species delimitation approaches that should be at least considered if some of the states want to be kept.

Specific

Line 15: I suggest checking the author guidelines and see if keywords need to be alphabetical

Line 21: “…with limited sampling…”; Taxonomic, population, or genetic/genomic sampling?

Line 39: As currently written, I don’t think the comma after cryptic species is necessary

Line 51: In cases like this, isn’t it better to write “Lei and Hanski (12), instead of just the number?

Line 58: missing a period at the end of the sentence.

Line 59: missing an apostrophe in worlds, since its not multiple worlds

Line 63-64: This sentence is a bit confusing. Cotesia has 300 named species where was Microgastrinae is 1500-2000, yes? I suggest revising to make it more clear

Line 64: The “the” starting the sentence can be deleted and just start with Cotesia

Line 66: how many is “few”?

Line 67: should Melitaeini be italicized here? The presence of an authority suggests yes

Line 70: comma between the two species can be removed. Has C. melitaearum been used previously, it doesn’t have an authority

Line 73: I think this is another case when providing the author names would be helpful; also, I don’t think “indeed” is necessary

Line 74: comma between species and are is not necessary

Line 78: how many are “few”? Please be explicit

Line 79-80: which two mito genes? Which nuclear locus? Please be explicit

Line 82: What values do the author consider “low bootstrap”? Under 70? Under 50?

Line 83: Why are the citations here formatted different than all others?

Line 85: The abstract says 22 species. What is the disconnect between 22 and 16?

Line 86: comma after butterflies is not necessary; how many markers? 10?

Line 97: missing a comma between the first two species

Line 98: Can abbreviate the genus name in the list after the first use, especially when there are over 8 for each genus

Line 105: similar to above, would be helpful to write out author names before (20)

Line 109-110: Is the second half of this sentence with “for the study of Cotesia wasps’ phylogeny and butterfly-host specialization in earlier studies….” Needed

Line 112-115: If the numbers are 1-9, they should be written out, such as one species, two species, etc. Cases of N=X can remain numbers

Line 116: Should fall be capiitalized?

Line 121: Is manipulated the best word here?

Line 139: I would make it clear in the Table legend that the GenBank numbers are for reference genomes

Line 149: Technically BLAST is a method not a verb; I suggest revising to “BLAST search” with the appropriate citations for BLAST

Line 163: should cite the previous studies here, especially since “…primer pairs designed and tested by previous studies…”

Line 164: Extra period after Table 2 here. Also, I thought Table 2 was for the genome assemblies used, not phylogenies?

Line 176-178: Since above there was clear distinction between mito and nuclear, I would make that clear in the table as well. How is it possible that two loci do not have amplicon size? If they amplified, it should be fairly clear what the size is

Line 183: What was the DNA concentration? Were they normalized across samples to ensure close to equal inputs?

Line 187: I would merge 1% gel electrophoresis and 30 minutes at 100 volts into the same sentence

Line 208: the comma between species is not needed here

Line 210: What settings for Muscle were used?

Line 211: Version of AliVew? Manually curated how?

Line 216-218: Were the topologies of these different subsets compared quantitatively, such as comparePhylo in Ape to get the number of shared nodes or splits?

Line 221: Why was BIC used for model selection instead of AIC given that likelihood and not Bayesian methods were used?

Line 224: With a relatively small data set, why not use 1000 regular bootstraps instead of ultrafast? Why not consider using site concordance factors or gene concordance factors as well, since these metrics give a good idea of support instead of just confidence

Line 235: Is there a table that includes gene accession numbers for each locus for each species? How many species only had 1 or 2 genes? Overall level of missingness in the data set?

Line 243: why include parsimony informative character stats if parsimony wasn’t used?

Line 246: Four clades were mentioned but only three support values were given. What happened to the missing value?

Line 282: statistics on how congruent the two trees were to each other?

Line 328: How was discordance characterized? The methods don’t suggest additional analyses for concordance/discordance.

Line 331-332: Was a coalescent based approach tried, such as Astral, with the individual genes?

Line 344: apostrophe isn’t needed here since multiple specimens are talked about, not possessive

Line 345: If species delimitation methods were not employed, even though they are in separate clades, can they truly be called different species?

Line 354: The term basal can be highly misleading since phylogenies can rotate. Can this be revised to be more explicit?

Line 355: What are the dates of divergence?

Line 363: again, basal can be misleading

Line 373-374: likely, yes, but there is not any expectation that every locus is going to show the exact same pattern. See Larson et al. 2026 in Evolution, discordance in the phylogenomics age

Line 375: “… differences between methods…” what differences in methods?

Line 380-381: This sentence seems to tear down the authors results. Do they not trust the results at all then?

Line 389: References from here on are not formatted the same as all the others

Reviewer #4: Dear authors,

I have reviewed your manuscript entitled "Progress in the molecular phylogeny of Cotesia acuminata and C. melitaearum cryptic species complexes." It is very nice work, although it requires some changes.

First of all, I would recommend changing the title, as the study is not only about Cotesia acuminata and C. melitaearum cryptic species complexes but also other species. I would also suggest removing "Progress in the" and simply using "Molecular phylogeny of..."

Second—and this is my major comment—you work with many cryptic species within C. acuminata and C. melitaearum, but no rigorous analysis (such as single-locus species delimitation, e.g., bPTP, GMYC, etc., or multilocus species delimitation, e.g., SODA) is presented. I believe that once you distinguish cryptic species, a statistical approach such as delimitation needs to be applied. I would also expect you to calculate genetic distances for particular loci and compare them within this study and with other studies.

I did not find Figure 1 in the MS. If I just overlooked it, I apologize.

I also have quite a few minor comments and edits mentioned directly in the text.

In general, your study presents a very interesting topic worthy of publication. However, before final acceptance, I would like to see the above-mentioned additions to your manuscript.

With best regards,

Petr Janšta

6. PLOS authors have the option to publish the peer review history of their article (what does this mean?). If published, this will include your full peer review and any attached files.

Reviewer #1: No

Reviewer #2: No

Reviewer #3: No

Reviewer #4: No

---

## [Author Response · Author response to Decision Letter 1]

22 Apr 2026

Please also see Response to Reviewers.docx

Reviewer #1: The paper presents a multi-gene phylogeny for braconid wasps in genus Cotesia with improved resolution. It is suitable for publication in Plos One after a minor revision.

In addition to the changes made in the attached word document, I have the following five main concerns.

Response: We thank the reviewer #1 for their assessment of our work. We have now edited the text according to the reviewers’ suggestions (please see the track changes manuscript), and a point-by-point response to each comment can be found below.

1) the abbreviation agg. should not be italicized, as it is not part of the scientific name of the species. The paper that the authors cite for the use of this term (Kankare et al 2005) also does not italicize the term.

Response: All instances of the abbreviation agg have been changed to non-italicised.

2) Table 1 does not include GenBank numbers for each individual analysed, and it is unclear which gene was sequenced for which sample. From Table 1 it can be inferred that the authors had a total of about 105 individuals (plus some from NCBI). A new supplementary table is needed with a list of these 105 individuals, their collection data and GPS coordinates (even if approximate), followed by columns for each of the 10 genes and the GenBank accessions for each individual clearly indicated.

Response: Following the reviewer’s suggestion, we have added a table to the supplementary material (S1 file). This table includes the accession numbers for each individual and gene, the collection data, and approximate GPS coordinates.

3) The authors mention sympatric occurrence of some of the Cotesia species, however without a map it is difficult to understand the extent of such sympatry. A new separate figure is needed with a map marking the geographic locations of the studied samples. Different symbols and colors should be used for the dots on the map to show: a) samples belonging to the four main clades, and b) whether or not they parasitize Melitaeini.

Response: We have created a figure with maps, which include the sampling locations of the specimens. Points are colour coded by clade and the shapes refer to the Lepidoptera host (i.e. Melitaeini or not). Please see Fig 1.

4) It is unclear how the authors decided to separate and highlight their four clades. For example, Clades C and D could be considered a single clade … I suggest that the authors conduct a basic Species Delimitation analysis (such as ASAP or PTP) or something similar, at least on their final phylogeny, to support their clade/species designations and conclusions.

Response: Thanks for the suggestion, we have now ran a bPTP analysis and BPP as recommended by other reviewers. This analysis shows the support for the four clades and cryptic species as previously described in the ms.

5) The authors make no suggestions to improve the systematics of Cotesia. Could the four main clades perhaps be considered subgenera? or species-groups? I suggest the authors to include at least a paragraph about this in their discussion.

Response: We now have changed the clades from A-D to represent the species groups instead, additionally, in the discussion we state “Our study questions the need to revise the taxonomy of Cotesia to highlight the cryptic diversity, and the possible introduction of new sub-genera by taxonomists.” although we do not feel qualified as taxonomists to describe new sub-genera.

#######################

Reviewer #2: The authors undertake a study to investigate phylogenetic relationships among Cotesia species parasitizing the caterpillars of Melitaeini butterflies. They do this using 102 fresh samples of Cotesia combined with and eight species for which data was available on GenBank. Their outgroup consisted of two Microplitis species. They generated data from 10 gene fragments (8 nuclear, 2 mitochondrial) for their taxa and analyzed them in a concatenated framework and by individual gene using maximum likelihood.

The authors show the Cotesia associated with Meliteaini are non-monophyletic and the existence of multiple forms within the currently recognized species C. acuminata and C. melitaerum.

Response: We thank Reviewer #2 for their suggestions. We have taken those into consideration and outline the changes in our responses below.

My main critique of the study is the limited approach to phylogenetic analysis and the unsupported statements regarding the production of a “robust” phylogenetic framework.

The authors only use a single analytical approach to phylogenetic inference – maximum likelihood (ML) as implemented in IQ-Tree. While the authors do investigate the influence of combining mitochondrial genes with nuclear genes, and the results of individual gene analyses, these can merely identify contention within the data themselves rather than the effects of different analytical approaches. Most modern phylogenetic studies at least incorporate some form of coalescent approach in order to account for gene tree/species tree discordance and incomplete lineage sorting directly. The authors identify the presence of gene tree discordance through analysis of their individual loci. However, they do not produce a coalescent-based tree to help resolve this. A common software package for this is ASTRAL (https://github.com/smirarab/ASTRAL) which makes use of summary coalescence – analyzing a priori produced gene trees to infer a species tree. The authors have already done most the work producing individual ML trees and would simply have to provide these to ASTRAL to produce a species tree. Additionally, given the highly tractable size of their data, they could also perform Bayesian inference (BI) using MrBayes (https://nbisweden.github.io/MrBayes/) or Beast (https://www.beast2.org/tutorials/).

A highly robust phylogenetic framework would be obtained from reciprocal illumination of results generated using different analytical approaches such as ML, BI, and ASTRAL. If these three fundamentally different approaches produce similar trees, then the data and inferred evolutionary history is robust to analytical approach.

Finally, the results the authors did obtain do not provide robust resolution across their datasets. An ultrafast bootstrap value of ≥95 is considered strong support (see reference 38), anything below this is not strongly supported. Figure 1, Clade (B(C+D)) shows low bootstrap support 77 (the nest node is illegible), Figure 2, the same Clade (B(C+D)) has similarly low bootstrap support – 82, then 87. Finally, figure 3, the analysis of mitochondrial genes alone recovers a different topology where B moves out and A moves in to clade. Bootstrap value is notably missing for this node in Figure 3. Combined these suggest that both within analysis support for much of their recovered topology is not strong, and the same topologies are not recovered across all datasets. So, while the recovery of species-group clades is consistent, relationship among said clades and thus the backbone of the phylogeny is not robust.

I recommend the authors include additional analytical approaches, at the minimum ASTRAL. Ideally, BI and ASTRAL.

Response: Thank you for the suggestions. We have added an analysis using ASTRAL to our study. We have edited the method, results and discussion sections to include this work (see methods (pages 13-14) and results (page 20)). In brief, ASTRAL recovered the same major clades as our original ML tree. However, with 10 loci, the signal was too weak to resolve within clade relationships. As similar results were obtained from both analyses, we did not find the need to run a first type of analysis through BI, which is likely to produce the same output based on our dataset. In the future, genomic data will be best analysed using BI, especially in the context of estimating the divergence time between species, clades, or maybe subgenera.

The authors indicate they suspect cryptic species, they could also perform a species delimitation analysis using BPP (https://github.com/bpp/bpp) and their mitochondrial data as well as their preferred tree as a guide tree to see how many of these potential cryptic species-level units are supported in a multispecies coalescent framework. Currently they have identified consistent splits in ML trees, but the nature of which of these splits are species-level is unclear, particularly for C. melitaearum which shows less structure than C. acuminata. Species delimitation may support C. melitaerum samples as being part of a single species relative to other taxa in the tree like C. acuminata.

Response: Thank you for the suggestion, we have included BPP analysis along with bPTP to test species delimitation, both of which support the presence of cryptic species as described previously (Kankare et al 2005). See results (page 21)

On lines 90 – 92 the authors state they want to resolve basal nodes of Cotesia, suggesting the broader phylogeny is of interest to them. However, based on the rest of the study it really seems the true focus is that discussed in 85 – 87. Therefore, I suggest removing this line as the broader phylogeny of Cotesia was really not a focus.

Response: We have removed this line

The authors fail to reference Michel-Salzat & Whitfield 2004 which appears to be an important study on the broader Cotesia phylogeny. This should be referenced.

Michel-Salzat, A. and Whitfield, J.B. (2004), Preliminary evolutionary relationships within the parasitoid wasp genus Cotesia (Hymenoptera: Braconidae: Microgastrinae): combined analysis of four genes. Systematic Entomology, 29: 371-382. https://doi.org/10.1111/j.0307-6970.2004.00246.x

Response: We thank the reviewer for recommending this article, we have now included the reference in our article.

Finally, a note that the discussion should probably be narrative and not include bullet points. There are also a couple issues with incorrectly formatted references (Line 56 & 83).

Response: Thank you for bringing up these mistakes. We have corrected the formatting issues following the instructions from PLOS ONE. Additionally, bullet points are now only present in the results section.

#######################

Reviewer #3: General

The submitted manuscript by Schach et al. uses a phylogenetics approach to investigate species complex relationships in a parasitoid wasp group. The analyses use 10 loci (eight nuclear and two mitochondrial) and a concatenated maximum likelihood approach. The results showed that the parasitoid wasp lineages were paraphyletic, resulting from numerous shifts in hosts.

Many recent/current phylogenomic studies utilize Illumina NGS data to isolate hundreds to thousands of loci. While that is common, not every study needs that level of data, especially projects that are long-standing or have a small budget. I am not going to ask for that here, since that seems unrealistic. However, I would say that many of the current analysis approaches used in those study should be at least considered here. Those include a coalescent-based approach such as Astral. Ten loci is not very many, yet there may be enough signal there. The concatenated approach is likely the better option, as shown here, but more work should be done to show concordance/discordance in the data used. Approaches such as PhyParts or gene concordance factors would show how many of the individual genes support different nodes in the concatenated tree. Bootstrap support, which tends to be inflated in phylogenomic data sets, is primarily a metric of confidence in the topology, not so much in the level of support of the data (either genes or sites) of the topology. In addition to bootstrap support, it would be useful to see how site concordance factors support the concatenated topology. [...] Overall, there is nothing wrong with the underlying data. This is a great foundation for a solid manuscript, but some reworking of analyses and revision of the main text is needed before publication. One specific example is the talk of species or cryptic species; a monophyletic group in a phylogenetic tree may not be enough evidence to assign species. There are numerous species delimitation approaches that should be at least considered if some of the states want to be kept.

Response: We thank reviewer #3 for taking the time to review this manuscript and for insightful suggestions, which have helped us improve the quality of this manuscript. Thank you also for their encouragements, which have supported the learning and motivation of all co-authors. We have included additional analyses, including gene concordance factors on the concatenated analyses, coalescent based approach with ASTRAL and species delimitation analyses with BPP and bPTP. We hope that these additional analyses will satisfy the curiosity of the reviewers.

I have a lot of specific comments to follow, many of these involve the presentation of the study. There are numerous grammatical or punctuation mistakes throughout. A careful revision is needed. Towards the end of the discussion the formatting of the intext citations completely changes from the rest of the manuscript.

Response: Thank you for pointing these out. We have reformatted the reference list and corrected typos highlighted by the reviewers.

I also have many comments involving being explicit. Instead of saying “a few”, say how many. When saying that mito and nuclear loci have been used previously, say which ones. These things will help nonexperts in parasitoid wasp evolution understand the manuscript.

Response: We have corrected this to explicitly state which loci have been used.

Line 15: I suggest checking the author guidelines and see if keywords need to be alphabetical

Response: There are no particular guidelines for keywords for the PLoS ONE journal, as keywords are optional for this journal. However, we place them in alphabetical order.

Line 21: “…with limited sampling…”; Taxonomic, population, or genetic/genomic sampling?

Response: This has been updated to include both taxonomic and genetic sampling

Line 39: As currently written, I don’t think the comma after cryptic species is necessary

Response: The comma has been removed

Line 51: In cases like this, isn’t it better to write “Lei and Hanski (12), instead of just the number?

Response: This sentence has been changed to: For example, it has been shown that the specialist species Cotesia melitaearum (Wilkinson), parasitizes only 10% of Melitaea cinxia (Linnaeus) caterpillars but can still cause localized extinctions within the butterfly host metapopulation [12].

Line 58: missing a period at the end of the sentence.

Response: A period has been added

Line 59: missing an apostrophe in worlds, since its not multiple worlds

Response: An apostrophe added

Line 63-64: This sentence is a bit confusing. Cotesia has 300 named species where was Microgastrinae is 1500-2000, yes? I suggest revising to make it more clear

Response: Cotesia has 1500-2000 estimated species with over 300 of them being already named. The sentence has been changed to: Within the Braconidae, the subfamily Microgastrinae is the most diverse. For example, the genus Cotesia (Cameron) contains an estimated 1500-2000 species worldwide [16] with over 300 of which have been named as species [17].

Line 64: The “the” starting the sentence can be deleted and just start with Cotesia

Response: “The” has been removed

Line 66: how many is “few”?

Response: We have changed few to two making it more explicit: Cotesia wasps usually exhibit narrow host ranges, and closely related species often differ in their host species preferences, attacking one or two closely related lepidopteran host species [18]

& Line 78: how many are “few”? Please be explicit

Response: The sentence have been updated to be more specific Currently, the phylogeny of the Catalonian C. acuminata complex and C. melitaearum complex are based on 12 microsatellite loci, while a wider Eurasian phylogeny of these groups combined two mitochondrial (COI and ND1) and one nuclear (ITS2) gene sequences [19].

Line 67: should

---

## [Editor Report · Decision Letter 1]

24 Apr 2026

Progress in the molecular phylogeny of Cotesia acuminata and C. melitaearum cryptic species complexes

PONE-D-26-02522R1

Dear Dr. Schach,

We’re pleased to inform you that your manuscript has been judged scientifically suitable for publication and will be formally accepted for publication once it meets all outstanding technical requirements.

Kind regards,

Petr Heneberg

Academic Editor

PLOS One
---

## [Editor Report · Acceptance letter]

PONE-D-26-02522R1

PLOS One

Dear Dr. Schach,

I'm pleased to inform you that your manuscript has been deemed suitable for publication in PLOS One. Congratulations! Your manuscript is now being handed over to our production team.

Kind regards,

on behalf of

Dr. Petr Heneberg

Academic Editor

PLOS One